# Comparison of Coupled Electrochemical and Thermal Modelling Strategies of 18650 Li-Ion Batteries in Finite Element Analysis—A Review

**DOI:** 10.3390/ma16247613

**Published:** 2023-12-12

**Authors:** Bence Csomós, Szabolcs Kocsis Szürke, Dénes Fodor

**Affiliations:** 1Research and Development Center of Technical Sciences, University of Pannonia, H-8200 Veszprem, Hungary; 2E-Mobility Research Center, Department of Power Electronics and Electric Drives, Audi Hungaria Faculty of Automotive Engineering, Széchenyi István University, H-9026 Gyor, Hungary; fodor.denes@ga.sze.hu; 3Department of Road and Rail Vehicles, Audi Hungaria Faculty of Automotive Engineering, Széchenyi István University, H-9026 Gyor, Hungary; kocsis.szabolcs@sze.hu

**Keywords:** Li-ion battery, coupled thermo-electrochemical modelling, thermal imaging, finite element analysis, temperature distribution

## Abstract

The specificities of temperature-dependent electrochemical modelling strategies of 18650 Li-ion batteries were investigated in pseudo-2D, 2D and 3D domains using finite element analysis. Emphasis was placed on exploring the challenges associated with the geometric representation of the batteries in each domain, as well as analysing the performance of coupled thermal-electrochemical models. The results of the simulations were compared with real reference measurements, where temperature data were collected using temperature sensors and a thermal camera. It was highlighted that the spiral geometry provides the most realistic results in terms of the temperature distribution, as its layered structure allows for a detailed realisation of the radial heat transfer within the cell. On the other hand, the 3D-lumped thermal model is able to recover the temperature distribution in the axial direction of the cell and to reveal the influence of the cell cap and the cell wall on the thermal behaviour of the cell. The effect of cooling is an important factor that can be introduced in the models as a boundary condition by heat convection or heat flux. It has been shown that both regulated and unregulated (i.e., natural) cooling conditions can be achieved using an appropriate choice of the rate and type of cooling applied.

## 1. Introduction

Li-ion technology is considered the most dominant electrochemical energy storage for portable hybrid electric vehicles (HEVs) and electric vehicles (EVs) [1]. Charging and discharging Li-ion batteries involves the transport of lithium ions among porous electrodes immersed in electrolytes, resulting in energy storage or release [2]. The electrodes are composed of a characteristic metal matrix. The anode is usually made of graphite, and the cathode is often a lithium–nickel–manganese–cobalt oxide (NMC) or lithium–ferric phosphate (LFP) matrix, with the choice of additives being left to the battery manufacturer. The widely used electrolyte is made from an organic solvent that typically forms a LiPF_6_ solution in commercially available batteries. A battery based on lithium-ion technology is sensitive to ambient and operating temperatures, which significantly affect the actual performance and safety of the cell [3]. Although it is considered a safer environment, low temperatures reduce the overall performance of the cell, notably the charge and discharge rates, and excessive Li surface formation can occur at too low temperatures [4]. High temperature is also a limiting factor, as solid electrolyte interphase (SEI) formation and electrolyte decomposition at high temperatures can cause thermal runaway [5].

To ensure safety and long life, automotive, portable and stationary applications using lithium-ion batteries are equipped with a battery management system (BMS) [6]. The accuracy of the thermal model of the monitoring system built into the BMS affects the effectiveness of state estimation, such as state-of-charge (SOC) or state-of-health (SOH) estimation and cell failure prevention. Therefore, several improved thermal models for battery condition estimation and numerical models for off-line thermal analysis of batteries have been published in recent years. The former uses a priori thermal equations to fine-tune the parameters [7] while the latter aims to reveal the spatial distribution of heat losses and temperature changes during charging and discharging. It has also been reported [8,9,10,11] that off-line evaluation of the thermal behaviour of batteries helps to incorporate thermal equations into BMS [12] and to optimise battery cooling strategies [13,14].

The results of battery thermal modelling achieved by finite element analysis (FEA) are manifold. In presenting the relevant results achieved recently, we believe that the most consistent logical order is achieved by grouping them by modelling dimensions and describing them as such. Within a group, a chronological order was established among the works presented. Before moving on to this grouping, however, it is worth noting that commercially available batteries are typically cylindrical (e.g., 18650), prismatic (e.g., pouch) or coin shaped (e.g., CR2032), and recent studies have taken one of these types as the basis for their analysis.

The fundamentals of the mathematical background of coupled thermo-electrochemical modelling in numerical simulations were laid down by Gu and Wang et al. [15,16,17].

For thermal studies using a 1D or pseudo-2D electrochemical model, Wang et al. [18] developed a 1D coupled thermo-electrochemical Li-ion battery model in 2013, showing how 1D models can be used effectively to estimate the temperature inside 18650 batteries, which is difficult to measure in practice. Saw et al. [19] analysed a LFP 18650 cell in a finite element environment where a 1D DFN (Doyle–Fuller–Newman) model and a 3D averaged thermal model were used. In 2014, Zhang et al. [20] investigated which spiral cells can neglect the thermal propagation in the direction of the axis and electrode thickness. They used a p2D electrochemical model to describe the thermal evolution, coupled with a simplified one-dimensional thermal model. They investigated how the cell diameter affects the accuracy of the model, for example for 18650 and 26650 cylindrical batteries. Baba et al. [21] presented a method to use an enhanced DFN model in 1D while the thermal model is in 3D. In 2016, Jokar et al. [22] summarised which are the most common coupled thermo-electrochemical models. On the basis of their research results, it was found that the most popular model describing the electrochemistry of batteries is the DFN model in pseudo-2D, with the use of a single particle model as a key computational resource reduction. In 2018, Sambegoro et al. [23] investigated the thermal behaviour of a prismatic LMO (lithium manganese oxide) battery using FEA in Ansys software (https://www.ansys.com/applications/battery, access date: 27 November 2023) and successfully detected the temperature distribution at the battery’s surface. Tran et al. [24] successfully created a computationally optimised p2D electrochemical 2D thermal model using Padé simplification. Their work was based on an 18650 cylindrical cell. Their publication does not address the visualisation of the internal or surface temperature distributions of the cell. At the same time, Raihan et al. [25] similarly investigated the finite element applicability of a p2D electrochemical 2D thermal coupled model for the power density optimisation of silicon and graphite anodes. In 2019, Chiew et al. [26] constructed a p2D electrochemical 2D thermal model of a LFP/graphite battery that fits well to charge cell voltage and charge cell temperature measurements. In their work, they also investigated the cell surface temperature characteristics as a function of charge under different load currents using thermal imaging. In 2020, Wang et al. [27] coupled a p2D electrochemical p2D thermal model with the 18650 NMC cell, which also considered current collectors. The cell, with its helical structure, was modelled in its extended (i.e., planar) state, in a complete charge–discharge–relaxation cycle. The cell was heated to approximately 6 °C in 800 s at a load current of 2 C and cooled down to the initial temperature in 2000 s.

For thermal studies using a 2D electrochemical model, comprehensive summaries of the various multiscale and multidimensional modelling concepts were published by Xu et al. [28] and Ye et al. [29], which can be considered very detailed summaries of the FEA of batteries in 2D. Numerous authors have investigated cylindrical cells in 2D to model cell failures [30,31] and optimise the cooling [32]. On the other hand, some other researchers have studied pouch cells in 2D [33] and 3D [34,35,36] finite element environments. Tong et al. [37] analysed cooling optimisation techniques in 3D.

For thermal studies using a 3D electrochemical model, Robinson et al. [38] published a comprehensive study on the nonuniform temperature distribution of lithium cells in 2018, in which they demonstrated the fundamental thermal behaviour of an 18650 cylindrical-type cell based on thermal imaging and X-ray microtomography. In addition, they highlighted critical parts of the 3D-formed compartment of the 18650 cell that are subject to excessive heating. Later, Bolsinger et al. [39] also pointed out the excessive temperature rise in the negative terminal of an 18650 cell based on thermal imaging measurements and 3D modelling. Jindal et al. [40] recently published a study on the detection of thermal runaway in electric vehicle (EV) battery packs using 3D numerical simulations. The effects of ageing on NMC cells and battery packs have been investigated in both pseudo-2D and 3D dimensions using FEA [41]. In addition, the temperature distribution of a cylindrical battery exposed to 1 C–4 C load currents is shown in 3D. Li et al. [42] built a coupled thermo-electrochemical model of a prismatic cell in the Comsol 5.4 finite element environment to evaluate the effects of meshing and geometric configurations. There were some attempts to reduce the computational complexity of 3D multiscale models by model reduction, which is based on an interconnection of elementary cells [43,44]. The large-format, spirally wound cells can be meshed using quadratic mapped elements, which can reduce the overall DoF (degrees of freedom) and increase the speed of computation [45].

## 2. Scope of the Paper

Although the efficiency and accuracy of the models reported in the previous papers are individually defined, they are not comparable, as most of them have been tested in different dimensions, under different operating conditions and with different battery structures. The aim of this paper was to reproduce the popular models for 18650 cells under the same operating conditions and to compare their simulation results with each other and with reference measurements.

Five different realistic models were chosen and implemented. The characteristics of the simulated terminal voltage and temperature were compared with each other and with measured (i.e., reference) data. In these models, regulated (i.e., forced) and unregulated (i.e., natural) cooling scenarios were realised, whose performances were benchmarked using three different types of reference measurements. The pitfalls of modelling in an axisymmetric domain are also detailed.

## 3. Materials and Methods

### 3.1. Governing Electrochemical Equations Used in FEA

Regardless of the dimension in which the cell is modelled, the DFN model is the most widely used approach to provide a good representation of battery dynamics in FEA [46]. The DFN model is considered accurate up to cell currents of magnitude 1 C, and it is not recommended for modelling currents higher than this [47]. The DFN model is based on the mass and charge conservation equations applied to solid and liquid components, such as electrodes, separators, current collectors, and electrolytes. The DFN modelling strategy uses the theory of porous electrodes [48], whereby the particles are assumed to be spherical and the material properties of the electrodes are defined parametrically. The electrolyte is assumed to be composed of a binary salt and a polymer solvent using concentrated solution theory [49]. We implemented the porosity of the electrodes and the separator parametrically via the volume fraction of the liquid and solid phases in the corresponding component of the cell. All relevant DFN parameters and their relationships are presented in Figure 1.

The mathematical apparatus of the DFN model used in the FEA is summarised in Table 1.

Equations (1)–(5) are the main governing equations of the DFN battery model, and they are clearly dependent on the absolute temperature, T. The electrochemical system they define is temperature dependent, and the parameters in the equations, such as $D_{s,eff}$, $D_{l,eff}$, $\kappa_{l}$, $f_{+-}$ and $t_{+}^{0}$, are themselves functions of several factors, such as the concentration, temperature, SOC and SOH. The vital parameters of a DFN model implemented in an FEA to describe the main Li-ion battery dynamics are the concentration, temperature and the open circuit potential (OCP). In order to exploit the dynamic behaviour of the DFN model, it is necessary to implement the functions $D_{s,eff}\left(c_{s},T\right)$, $D_{l,eff}\left(c_{l},T\right)$, $\kappa_{l}\left(c_{l},T\right)$, $f_{+-}\left(c_{l},T\right)$ and $t_{+}^{0}\left(c_{l},T\right)$, as well as *OCP*(*SOC*).

### 3.2. Governing Thermal Equations Used in FEA

The generation of heat in a battery is due to thermodynamic and ohmic losses. The total amount of heat generated during the operation, which causes its temperature to rise, consists of the sum of ohmic, reversible and irreversible heat losses [21]. Mathematically, it can be written in the following form, in general:(6)$$Q_{tot}=Q_{Ohmic}+Q_{rev}+Q_{irrev}$$

Each term in Equation (6) can be expressed as shown in Table 2.

In order to obtain a more realistic thermal description of the electrodes and the separator, it was found advantageous to model them as a porous medium. In this case, the thermal conductivity of the electrolyte inside the electrodes can be parameterised more accurately; otherwise, either the solid or the liquid component must be omitted from the electrode model. In porous media, heat transfer can be introduced by expressing it in terms of $\lambda_{eff}$ according to:(11)$$\lambda_{eff}=\epsilon_{solid}\lambda_{solid}+\epsilon_{fluid}\lambda_{fluid}\;\;$$
where $\epsilon_{solid}$ denotes the volume fraction of the solid, $\epsilon_{fluid}$ stands for the volume fraction of the void in the electrode, and $\lambda_{solid}$ and $\lambda_{fluid}$ are the thermal conductivity of the solid and fluid, respectively.

### 3.3. Consideration of Modelling Domains

As mentioned in the Section 1, the electrochemical and thermal behaviours of batteries can be studied in 1, 2 and 3D. In a coupled model set-up, the electrochemical and thermal model parts can be in the same dimension but not necessarily. Figure 2 shows the typical design of the cell geometry and choice of boundary conditions implemented in 1, 2 and 3D.

In general, 1D is the simplest and most convenient domain for coupled electrochemical and thermal models, in which the two electrodes and separator of the battery are represented only by their thickness along the *x*-axis.

In other words, the spirally wound cell is modelled as unwound in both the electrochemical and thermal model sections. The unwound sandwich cell model, consisting of three to five layers depending on the current collector modelling, results in a so-called “unit cell” configuration. In this case, the cell is scaled by the cross-sectional area of the separator, $A_{sep}$, which is perpendicular to the *x*-axis and equal to the product of the electrode length and height, which are pseudo-parameters.

In principle, p2D models can be used in all cases in which the spatial distribution of the battery housing, battery cap, multiple batteries in a battery bank is negligible, i.e., not a test criterion.

In a realistic thermal model, the cooling of the cell must also be taken into account, regardless of the modelling dimension. In principle, the cooling effect can be introduced in two ways: either by defining the heat flux at a specific point or boundary or by applying thermal convection. In the p2D domain, only the heat flux can be used as a boundary condition, which only needs to be set at the endpoint of the extended cell model where the battery house is localised. This implements a realistic case in which the battery can only dissipate heat and, thus, cool on its enclosure wall. A 2D or 3D thermal model should be coupled to the p2D electrochemical model if heating is to be investigated in individual assemblies or in the interaction of several cells within a module. A good example of the latter is the cooling of a battery module of 18,650 cells.

In 2D, as shown in Figure 2, the cell structure can be modelled with concentric or spiral layers, which provide a more sophisticated analysis than p2D models. Since the spiral is not symmetric in the longitudinal section, it can only be used for modelling in a top view, i.e., cross-sectional perspective. If the electrochemical model is represented in 2D, it is possible, for example, to model the ohmic resistance of the current collectors, which allows for a more detailed model of the heat distribution inside the cell. In the 2D electrochemical model, the resultant capacitance of the cell is determined by the thickness of the layers and the number of turns of the spiral, i.e., the total surface area of each layer—just as in reality. From the point of view of the thermal behaviour, the 2D model is suitable to study the temperature distribution inside the cell, for example, around the mandrel or around the casing wall. The maximum temperature slightly changes depending on whether electrolyte or air fills the gap between the mandrel and the innermost spiral layer and between the can wall and the outermost spiral layer.

In 2D, it is also theoretically possible to exploit geometric symmetries, which can speed up modelling and its numerical solution. In this case, each layer, corresponding, for example, to a current collector, electrode or separator, can be modelled by an elementary self-locking cylindrical unit. These independent elementary units must be combined to form a complete work cell. This interconnection can be achieved by conductive interconnections between the current collectors or by using boundary conditions to define the surface currents of each current collector.

Using the geometric layout shown in Figure 3 leads to a volume averaged heat transfer model in which the details between layers are neglected.

Even though the thermal model can be implemented as shown in Figure 3a, the concept of such a merged unit suffers from the problem that a reduced reaction area appears between the electrodes (highlighted in green in the 3D rotational subplot of Figure 3a). The reduced reaction area is due to the reduced number of layers, i.e., the reduced surface area through which the ionic current of the cell can flow. The reduction in the reaction area can be traced in the reduced overall capacity of the cell.

The approach shown in Figure 3b has some significant drawbacks that make modelling in this way challenging. First, the additional paths used introduce extra heat conduction paths that distort the actual heat distribution. In addition, the distortion will be huge, since the interconnections are disks rather than prisms due to the 2D rotational symmetry. On the other hand, the use of surface currents at the layer boundaries as boundary conditions is also not reasonable, since the currents in the cross-sectional areas of the inner layers (highlighted in green in Figure 4) are unknown. Because of the problems mentioned above, this concept is generally not favoured.

The use of spiral geometry in 3D, both in the electrochemical and thermal models, seems possible. However, the 3D spiral shape introduces a spline-based extended object where the thickness of the electrode, current collector and separator can be measured to within a few tenths of a micrometre, while the height of the cell is on the order of millimetres. Such a difference in magnitude causes networking problems and even an optimal mesh configuration can result in a huge number of degrees of freedom (DoF) [29]. In other words, although the 3D spiral basis is promising because of the realism of the model, working with 3D spiral geometry is very resource-intensive and generally unmanageable in an FEA.

Another, simplified approach is to omit electrochemically active battery components, such as current collectors, electrodes and separators, from the battery’s geometry, resulting in a single, simplified domain. This concept is illustrated in Figure 2 (3D).

This configuration does not allow for the DFN model to be implemented, and only the thermal model can be handled in this way. The DFN model must be in a 1D or 2D format. If the DFN model is in p2D and the thermal model is in 3D, a lumped parameter model is obtained. In this case, the volumetric heat losses estimated by the p2D DFN model are scaled to the 3D thermal model. Mathematically, this can be expressed as follows:(12)$$Q_{tot,3D}=Q_{tot,p2D}\frac{L_{an}+L_{sep}+L_{cat}}{L_{cell}}\frac{\left({\left(R_{cell}-d_{can}\right)}^{2}-r_{cell}^{2}\right)\left(H_{can}-2d_{can}\right)}{\left(R_{cell}^{2}-r_{cell}^{2}\right)H_{can}}\;\;$$
where $L_{cell}$ denotes the total thickness of the unity cell including the two current collectors,$\;L_{an}$ stands for the thickness of the anode, $L_{sep}$ represents the thickness of the separator, $L_{cat}$ refers to the thickness of the cathode, $R_{cell}=D_{cell}/2$ is the outer radius of the cell, $d_{can}$ denotes the wall thickness of the can, $r_{cell}=d_{cell}/2$ stands for the radius of the mandrel and $H_{can}$ represents the total height of the cell.

Cooling can be introduced by the boundary heat flux on the external surfaces of the container or by heat convection with inlet and outlet boundary conditions.

### 3.4. Pros and Cons of the Models

In this paper, all of the models discussed earlier were developed, and the results were compared with thermal imaging and temperature sensor-based reference measurements. The pairs of models used in the comparison are summarised in Table 3.

## 4. Experimental

The battery models were based on the parameters of a commercially available Samsung ICR18650-26J 2.6 Ah cylindrical cell. According to Sommerville et al. [50], it consists of a NMC cathode and a MesoCarbon MicroBead (MCMB) graphite anode. The electrochemical parameters used (e.g., electrode porosities, particle sizes and Bruggeman coefficients) were adopted from Carelli et al. [51]. The separator porosity was taken from Tiedemann et al. [52]. It was assumed that the cell consisted of a conventional LiPF_6_ electrolyte in an organic solvent, that is, LiPF_6_ in 3:7 *v*/*v* EC:EMC. Although the mixture and volume ratios of the alkyl carbonate solvents affect the operating range and conductivity of the electrolyte, the initial salt concentration ($c_{l}$) of 1 mol/dm^3^ and transference number ($t^{+}$) of 0.363 are similar in these types of electrolytes, according to Valoen et al. [53]. The Li diffusion coefficient and the ionic conductivity of the electrolyte change with the electrolyte salt concentration; therefore, their values were given by functions. The maximum Li concentrations in the cathode and anode were chosen to be 49,000 mole/m^3^ and 31,507 mole/m^3^, respectively. These are material constraints and depend on the composition of the electrodes. The reference exchange current densities of the anode ($i_{0,an}$) and cathode ($i_{0,cat}$), which were taken from a previous paper [54], were 0.94 A/m^2^ and 0.40 A/m^2^, respectively. The thermal parameters of the components of the cell, such as the thermal conductivity (k), density (ρ) and specific heat capacity ($c_{p}$) at a constant pressure have been calculated and summarised by Maleki et al. [55] and Spinner et al. [56]. Data from Maleki’s paper were adopted. The main geometrical parameters (e.g., cell height and diameters, as well as the thicknesses of the electrode, separator and current collector) were measured using XT H 225 ST computed tomography (CT), and the results can be seen in Appendix A.

The real thermal behaviour of the cell was analysed by a Fluke Ti25 thermal imaging camera using an LM35 thermocouple and two LMT85LPG temperature sensors attached to the centre and the two terminals of the cell, respectively. The cell underwent two different types of measurements, that is, continuous and interrupted discharge.

During continuous discharge, the cell was kept under unregulated (i.e., natural) cooling conditions and discharged at a 1 C constant current (CC) rate for 480 s. The temperature was logged by the thermal imaging camera (Δ*Tcenter*, Δ*Tcell* and Δ*Tcap*), a thermocouple (Δ*Tsens*) and two temperature sensors (averaged in Δ*Tsens o.air*), simultaneously. The Δ*Tcenter* and Δ*Tcap* refer to a one-point-measured temperature at the centre and on the cap of the cell, respectively, while Δ*Tcap* measures an average temperature around the whole front surface of the cell. The temperature values from the two LMT85LPG sensors were averaged, and their averaged values are presented in the comparisons.

The continuous discharge was terminated after 480 s when the temperature change reduced to less than the 0.5 °C resolution of the thermocouple.

During the interrupted discharge, the cell was discharged at a 1 C constant current rate for 360 s, which was followed by 120 s of relaxation with no load. In this scenario, the cell was kept in an ESPEC LU-113 thermal chamber to provide regulated cooling conditions. The discharge time of 360 s was chosen to provide enough time to discharge the cell by 10% SOC. The temperature data acquired by the temperature sensor in the thermal chamber were labelled as Δ*Tsens c.chamber*. The measurement of the interrupted discharge was run three times to obtain temperature characteristics during regulated and unregulated (i.e., natural) cooling conditions within short (480 s) and long (3600 s) operating periods. In the case of unregulated conditions, the cell was placed and measured in both an insulator box and open-air conditions separately. These temperature data were named as Δ*Tsens box* and Δ*Tsens o.air*, respectively. All the set-ups are listed in Appendix A and shown in Figure 4.

Data acquisition was performed using customised LabVIEW 2013 software running on a NI PXI embedded controller. The simulations followed the same procedure as the real measurements to obtain comparable results. The simulation was run in COMSOL Multiphysics 5.5 on an Intel Core i7-8700 @ 3.2 Ghz CPU equipped with 16 GB DDR4 RAM. The geometrical and simulation parameters are summarised in Appendix A.

The thermal parameters were taken ”rom ’aleki et al.’s paper [55], and the material data were mostly collected from the COMSOL’s Material library. In the current work, the functions $D_{l}\left(c_{l},T\right)$, $\kappa_{l}\left(c_{l},T\right)$, $f_{\pm }\left(c_{l},T\right)$, $D_{s,an}\left(c_{l},T\right)$ and $OCP_{an}\left(SOC,T\right)$ were implemented using the following Arrhenius equations:(13)$$\kappa_{l}\left(c_{l},T\right)=\kappa_{l,0}\left(c_{l}\right)e^{\frac{4000}{8.314}\left(\frac{1}{298}-\frac{1}{T}\right)}\;\;$$
(14)$$f_{\pm }\left(c_{l},T\right)=f_{\pm ,0}\left(c_{l}\right)e^{-\frac{1000}{8.314}\left(\frac{1}{298}-\frac{1}{T}\right)}\;\;$$
(15)$$D_{l}\left(c_{l},T\right)=D_{l,0}\left(c_{l}\right)e^{\frac{17,200}{8.314}\left(\frac{1}{298}-\frac{1}{T}\right)}\;\;$$
(16)$$D_{s,an}\left(T\right)=1.45\cdot 10^{-13}e^{\frac{68,025.7}{8.314}\left(\frac{1}{298}-\frac{1}{T}\right)}\;\;$$
(17)$$OCP_{an}\left(SOC,T\right)=OCP_{an}\left(SOC\right)+\frac{dOCP_{an}\left(SOC\right)}{dT}\left(T-298\right)\;\;$$

The functions $\kappa_{l,0}$, $f_{\pm ,0}\left(c_{l}\right)$, $D_{l,0}\left(c_{l}\right)$, $OCP_{an}\left(SOC\right)$ and $dOCP_{an}\left(SOC\right)/dT$ can be found in lookup tables in the Appendix A. The $t_{0}^{+}$ values were assumed to be constant for both electrodes. The lookup tables, the equilibrium potentials of both electrodes and the constants in Equations (13)–(17) were adopted from COMSOL Multiphysics Model Library version 5.5. The characteristics of both of the electrodes are depicted in Appendix A.

## 5. Results and Discussion

### 5.1. Temperature and Terminal Voltage Characteristics

To obtain reference thermal data, continuous and intermittent discharge measurements were performed. The purpose of the reference measurements was to investigate the cell surface temperature, especially around the connectors, and to obtain high-resolution images of the local variations in the cell temperature. The side camera set-up was designed to detect temperature variations around the terminals. Both camera- and sensor-based data were collected to improve the accuracy of the measurements. The continuous discharge was used to analyse the thermal response of the models under normal operating conditions, while the interrupted discharge was used to study the transient thermal response of the system. First, during the continuous discharge, the temperature distribution across the cell was measured with a thermal camera from a frontal (Appendix A) and lateral view (Appendix A) to determine the local temperature differences at the cell surface.

Appendix A shows that the temperature distribution on the cell surface was not homogeneous. The upper and lower parts of the cell around the terminals were warmer than the centre until the cell reached a steady state (Appendix A/VIII). In Appendix A, these observations can be identified by observing a lower resultant temperature at the positive terminal than at the cell surface.

Furthermore, the electrochemical and thermal behaviours of the cell were analysed in the 1D, 2D and 3D domains using the geometries previously detailed in Section 1. The model pairings used in the simulations were as listed in Table 3.

In reality, in colder environments, a small air flow is likely to develop around a warm body, resulting in natural cooling. In our models, the effect of natural cooling was implemented using thermal convection and heat flux as the boundary conditions. The purpose of using two different types of boundary conditions was to analyse their different contributions to the cooling performance and cooling dependence of the thermal models. First, cooling was introduced by applying an arbitrarily chosen velocity of 0.1 m/s to the thermal convection, which seemed small enough to introduce a small breeze. The results and reference data are plotted against each other in Figure 5:

Figure 5 shows that in most models the average temperature difference of the cells fell in the range 1.5–2.5 K. All of the models showed a lower warming trend than the references’, which is addressed to an approximation error of the actual air flow built up around the cell in reality. The air flow of 0.1 m/s set in the simulations cools the cell better than the air in the laboratory. The effect of the location of the temperature sensors on the cell surface was also traced by the different characteristics of the reference temperature curves. The measured references show that the positive terminal maintained a lower temperature than the cell mantle, and the thermocouple data followed the trend produced by the camera-recorded data. The cell mantle approached thermal equilibrium after approximately 480 s, while the surface temperature of the fluid continued to rise slowly. The maximum temperature change during the test was 3.6 K.

Another way to introduce cooling is to use a heat flux. Ghisalberti et al. [57] used the empirical relationship between airflow and heat flux to choose a heat flux of 10 $W/m^{2}K$ for cooling, corresponding to an airflow of about 0.1 m/s. In this case, the results are similar to those shown in Figure 5 in terms of the final temperature reached during the simulation period. The results obtained using the heat flux are summarised in Figure 6.

On the basis of the results shown in Figure 6, the application of 10 $W/m^{2}K$ heat flux in order to model the cooling in this scenario is clearly a better approach. Most of the models followed the trends of the reference models quite well. Another important conclusion based on the good fit of the simulated and reference characteristics with each other is that the location of the temperature probes in the models are correct. Using the interrupted discharge, the transient response of the models was evaluated and compared with the reference. The results are shown in Figure 7.

Figure 7 shows that the cell cooled at a similar rate in most models. The terminal voltage response of the 2D-2D model was quite different than the measurements. This phenomenon was addressed to the approximation error of the number of layers in the cell sandwich, which results in a higher capacity and, therefore, a smaller voltage breakdown. During the 120 s relaxation, the average cell temperature decreased by 0.8 K, 0.51 K and 0.35 K in the p2D-p2D, p2D-2D and p2D-3D models, respectively. For these models, a comparison of the reference and simulated characteristics shows that the use of a thermal convection rate of 0.1 m/s is more suitable for modelling cooling in a climate chamber (i.e., forced cooling) than under free (i.e., natural) conditions.

Appendix A shows a segment of the potential distribution of the spirally wound, positive current collector in the 2D model.

The transient characteristics of each model changed slightly when a heat flux of 10 $W/m^{2}K$ was applied for cooling during the interrupted discharge. The results are presented in Figure 8.

The cooling effect noticeably reduced in this case, as can be seen from the increased temperatures and flat temperature curves in the relaxation phase. Comparison of the 2D-2D and p2D-2D models with measurements shows that the temperature responses of the 2D thermal models are the best fit to the reference temperature data. The p2D-2D and 2D-2D models are able to simulate outdoor and closed box conditions using a thermal flux of 10 $W/m^{2}K$. The p2D-p2D model reflects characteristics of cooling in a climate chamber while the characteristics of p2D-3D model falls in between.

To evaluate the model dynamics over long-term operation, all models were subjected to interrupted discharges up to 3600 s. The cyclic interruptions are promising for evaluating the extent of heating and cooling between each pulse. The discharge time of 360 s and the relaxation of 60 s were left unchanged for all interruptions to obtain results comparable to the short-term voltage and temperature characteristics. The completion criteria for the simulations was that either the simulation time reached the 3600 s time limit or the terminal voltage reached the 3 V safety voltage threshold. The voltage and temperature characteristics of each model during the 3600 s simulation are shown in Figure 9.

Figure 9 shows that the terminal voltages and temperature variations of all models follow the reference data reasonably well, suggesting that the thermal and electrochemical parameters are roughly the same as the real battery parameters. The largest deviation from the reference terminal voltage was approximately 80–100 mV between 420 s and 2945 s, as shown by the 2D electrochemical model. Furthermore, the 2D-2D model shows a voltage collapse earlier than expected, around 3200 s, which is due to the mismatch between the battery capacitance and the geometry implemented by the spiral-wound layers. The best fit to the reference voltage trend was obtained by the p2D-2D model, in which the fit error was negligible.

For the temperature characteristics, the best fit was obtained by the 2D-2D model with a maximum error of 0.7 K in approximately 1330 s. The good performance was due to the detailed internal structure of the jelly roll and the modelling of electrochemistry and heat transport in the same dimension. This finding may explain why the p2D models underperform in terms of temperature response.

The characteristics of the 2D-3D(ax) model presented earlier in Table 3 were evaluated separately, as it is a downscaled model and differs significantly from the others. For all model types, the cell was exposed to the same discharge current 1 C to ensure the same loading condition for all models. On the other hand, the amplitude of the current derived from the 1 C current always corresponded to the capacitance of the cell model. As the capacitance of the 2D-3D(ax) model decreased as detailed in Figure 3, the uniformly applied 1 C discharge current resulted in a much lower current than for the other models. As a result, the lower currents caused a smaller temperature increase, which significantly changed the characteristics of the 2D-3D(ax) model compared to the other models. The simulation results for the 2D-3D(ax) model are shown in Figure 10.

### 5.2. Sensitivity Analysis of the Model Parameters

During the model development, several key parameters were obtained from the literature and material library. Considering that these parameters may not be fully appropriate for the battery considered in this study, it was necessary to perform a sensitivity analysis on the relevant model parameters. Since tens of parameters are available in each modelling dimension, in order to reduce the computational demand, some of these parameters were selected, and the sensitivity analysis was performed on these parameters.

Electrode porosities, particle sizes, exchange current densities, ionic conductivity, activation dependency, diffusion coefficients, both in the electrolyte and the solid matrix, and heat transfer coefficients were the parameters for which the sensitivity of the model was tested. The output variable required to determine the sensitivity was the maximum temperature of the cell after immersion. The results of the sensitivity analysis can be seen in Figure 11, and the values are summarised in Appendix A.

Each thermo-electrochemical model is better adapted to different cooling scenarios in each dimension, i.e., when evaluating the results of the sensitivity test, the focus is not primarily on which model provides a higher maximum temperature (possibly closer to a chosen reference measurement) but on the temperature difference between the minimum and maximum parameter values. In other words, the smaller the slope of the sensitivity function, the less sensitive the model is to a given parameter change.

Figure 11 shows that the p2D-2D and the p2D-3D models were the most sensitive to changes in the given parameters, while the coupled 2D-2D model was the least sensitive.

### 5.3. Characteristic Temperature Distributions in the Internal Structure

The use of 2D and 3D thermal models allowed for the evaluation of the internal and surface temperatures of the cell. It also allowed for a qualitative comparison of the temperature distributions with real measurements. In the following, the results of the 2D and 2D axisymmetric thermal models are highlighted—the former to show the interlayer temperature and the latter to show, in detail, the effect of the cap and the metallic housing. The simulation results are shown in Figure 12:

It can be seen that the radial distribution of the temperature was altered by the different thermal resistances of the layers. It is important to note that the 2D spiral model has a pseudo-height equal to the $H_{can}$ value, and the cell can lose heat in this direction as well. The rate of heat convection and, hence, cooling depends on the rate of heat convection according to Equation (10).

The 3D model can be used to determine the effect of the air gap, insulators and metallic parts of the cap and bottom connector. The simulated segments are shown in Figure 13.

After 5 s, the temperature of the metal box increased and the box became warmer than the positive terminal next to the positive terminal, because the metal box distributed heat well in the box wall. In contrast, the cap slowed down the heat transfer due to the low thermal conductivity of the electrical insulators used and the air gap between the layers and the cap. In addition, the small air gap between the layers and the bottom terminal also reduced the cooling rate. After 100 s, the cell temperature approached a steady state, but the top terminal remained slightly cooler than the box. Along the longitudinal axis of the cell, the temperature distribution was fairly homogeneous, as the cell was heated as a single complete domain.

Note that neither laminar nor turbulent flow dynamics were used in these simulations to avoid overloading the model due to multiple couplings. Instead, we used a simple built-in convective airflow controlled by heat transfer physics.

As can be seen, there are several methods to develop an electrochemical and thermal model of an 18650 Li-ion cell. Despite the small differences in the electrochemical and thermal characteristics, by comparing the continuous and interrupted discharge results with references, it is shown that the real-scale models are all suitable for coupled electrochemical and thermal modelling. Bandhauer et al. [58] provided a comprehensive review of electrochemical and thermal modelling, in which their results showed that the average change in the cell temperature should converge to 2.5–3 K under steady-state conditions when a 1 C discharge current is applied. The reference data in Appendix A show that the cell surface around the cap and bottom terminal is strikingly more prone to heating than most of the surface. This inhomogeneity of the temperature distribution is consistent with the literature. For example, Waldmann et al. [59] discussed this phenomenon in detail. Three-dimensional models are able to capture this inhomogeneity and its characteristics, which are clearly shown in the three-dimensional simulation results shown in Figure 13. On the other hand, the use of 2D axisymmetric geometry is not suitable for coupled electrochemical and thermal models due to intrinsic geometric problems. Therefore, this approach is not recommended for coupled electrochemical and thermal modelling. Despite the fact that pseudo-2D-lumped models are simple and do not implement complex geometric structures, they are a very reasonable option for fast and resource-efficient simulations. The 2D electrochemical model performs well in transient studies and is the best fit to reference measurements. In all thermal models, the type of cooling (e.g., heat convection or heat flux) and the rate of heat convection have a large influence on the resulting cell temperature. Christensen et al. [60] published work in which forced convection with a fan and duct was used to eliminate cooling uncertainties. They used a heat flux of 12.5 $W/m^{2}K$ to introduce natural cooling in the thermal model, which is very similar to the 10 $W/m^{2}K$ value we used.

## 6. Conclusions

In this paper, all popular modelling strategies used in coupled electrochemical and thermal simulations in FEA are presented. On the basis of the comparison of the simulated and reference data, we show that the thermal effect of the cell in the electrochemical model can be efficiently realised using a pseudo-2D thermal model with a linear geometry of the cell, especially when the simulation needs to be simple and fast, for example, in cell ageing studies. The running time of p2D-p2D and p2D-2D models is approximately a half minute, while p2D-3D and 2D-2D models take about 1 minute and 2 and a half minutes, respectively.

The electrochemical model can be further improved by using 2D spiral geometry at the expense of the computational time, but this only yields notable improvements in transient simulations. In all fields, one of the most important factors that determine the performance of a model is the accuracy of the cross-sectional area of the layers, which contributes to the modelled capacitance of the cell. The main advantage of using 2D geometries comes from their helical layer structure, which results in a more realistic cross-sectional area compared to other approaches. The p2D-2D and p2D-3D couplings are less sensitive to parameter changes than the p2D-p2D and 2D models according to Figure 11. A 2D model can be created if the micro and macro dimensions of the cell are known, such as the thickness of the electrode, the separator and current collector, and the size of the canister. On the other hand, by using a 2D spiral geometry, attention must be paid to the number of layers in the spiral as this determines the overall capacity of the battery. The model usually works with ideal layers and, thus, an integer number of turns, but in reality the layers may be fragmented and contain imperfections that cause capacity loss.

The question of which model best and least approximates the reference data series cannot be answered unambiguously. As shown, the reference data series do not fit each other but show different trends depending on where the temperature is measured in the cell in reality, for example, at the cap or in the middle of the cell. The simulation results highlight that the consideration of the cooling capacity of the environment and the placement of the virtual thermometer probe in the model strongly influences to which reference data set the model output will be closer. Thus, a fit to either the temperature trend measured in the open-air or in the climate chamber may be correct, depending on which was the objective.

For measurements, it is recommended to avoid using thermocouples attached to any of the cell terminals, as they maintain a lower temperature than the base body and produce misleading readings that can cause the cell to overheat.

## Figures and Tables

**Figure 1 materials-16-07613-f001:**
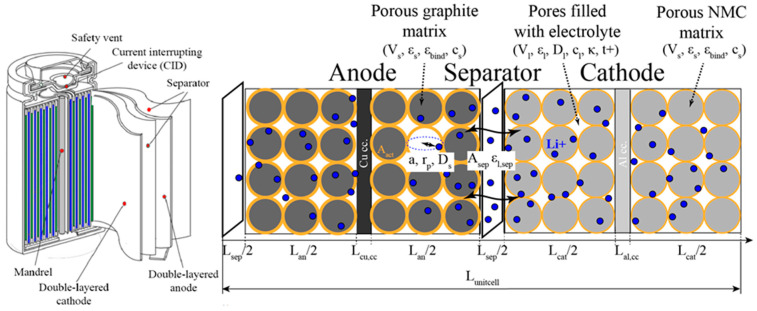
Schematic diagram of an 18650-type cylindrical Li-ion battery with double-coated electrodes.

**Figure 2 materials-16-07613-f002:**
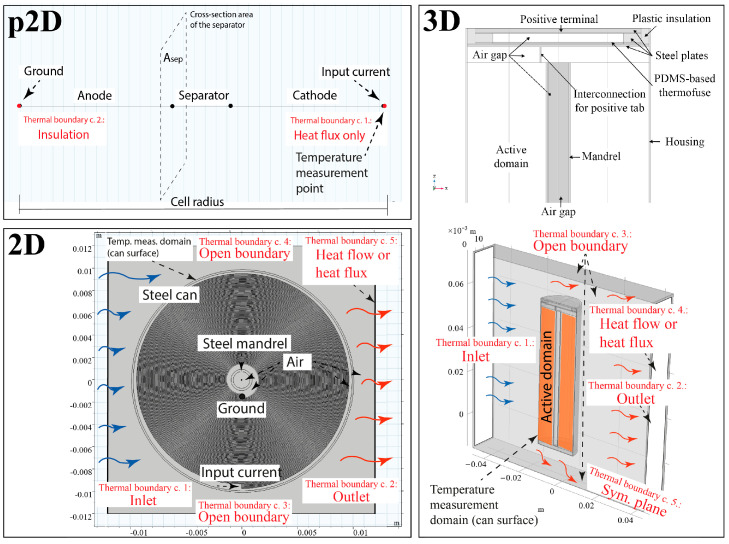
Typical model geometries and layouts in 1, 2 and 3 dimensions.

**Figure 3 materials-16-07613-f003:**
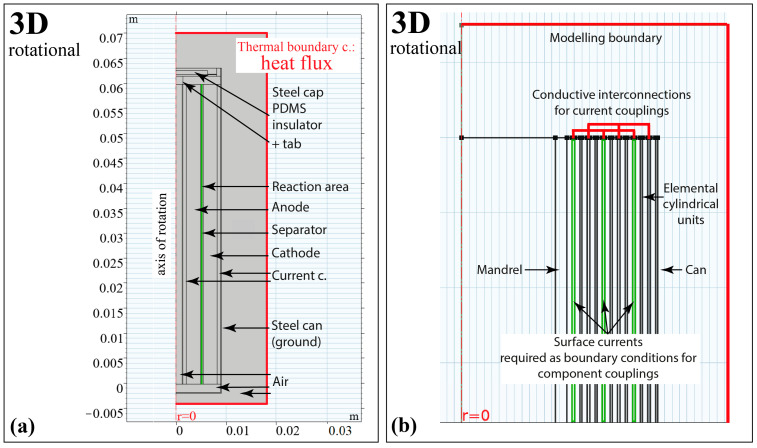
(**a**) Modelling rotationally symmetrical multilayered cell structures in 3D and its challenges; (**b**) if the jelly-roll layers should be indicated. In (**b**), the green zones indicate boundary conditions where surface currents must be set.

**Figure 4 materials-16-07613-f004:**
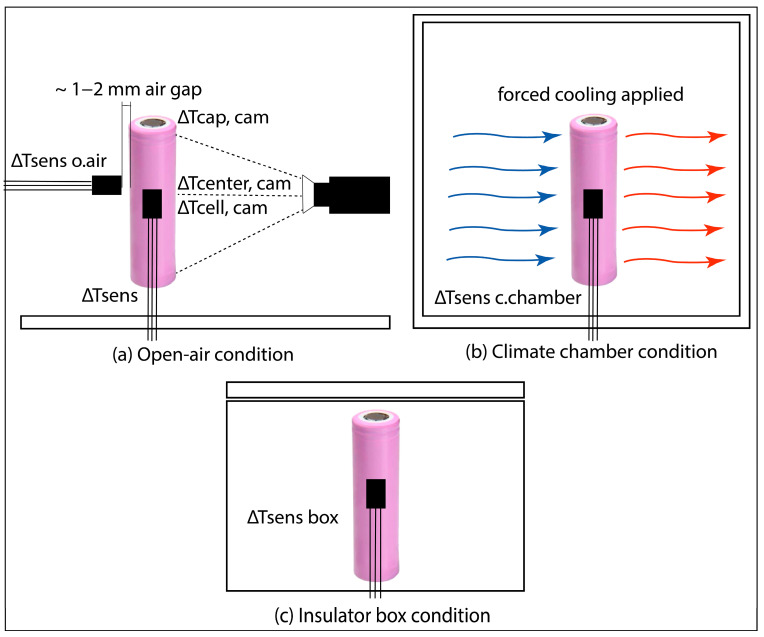
Experimental set-up for the (**a**) open-air, (**b**) climate chamber and (**c**) insulator box measurements.

**Figure 5 materials-16-07613-f005:**
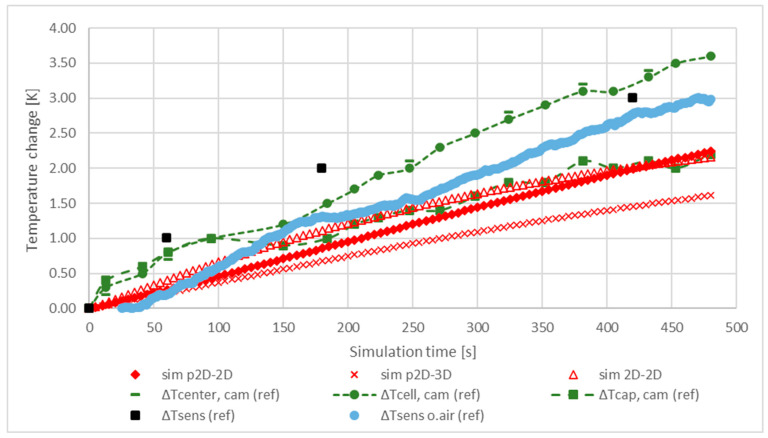
Comparison of the temperature characteristics of the inspected cell during a continuous discharge of 1 C for 480 s. Heat convection of 0.1 m/s was applied for cooling.

**Figure 6 materials-16-07613-f006:**
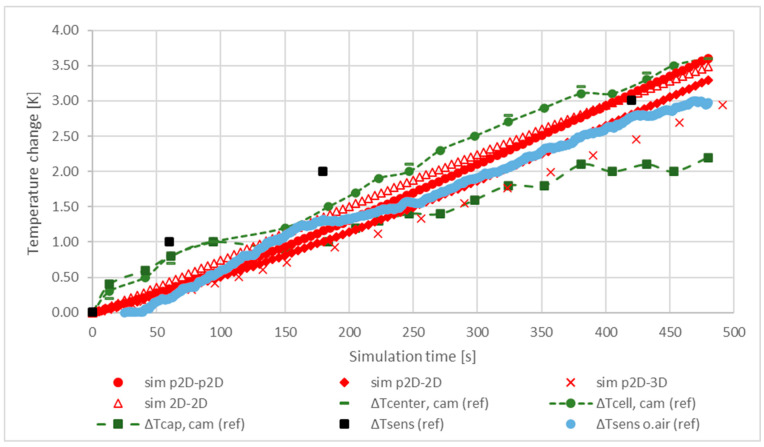
Comparison of the temperature characteristics of the 18650 Li-ion cell during a continuous discharge of 1 C for 480 s. A heat flux of 10 $W/m^{2}K$ was applied for cooling.

**Figure 7 materials-16-07613-f007:**
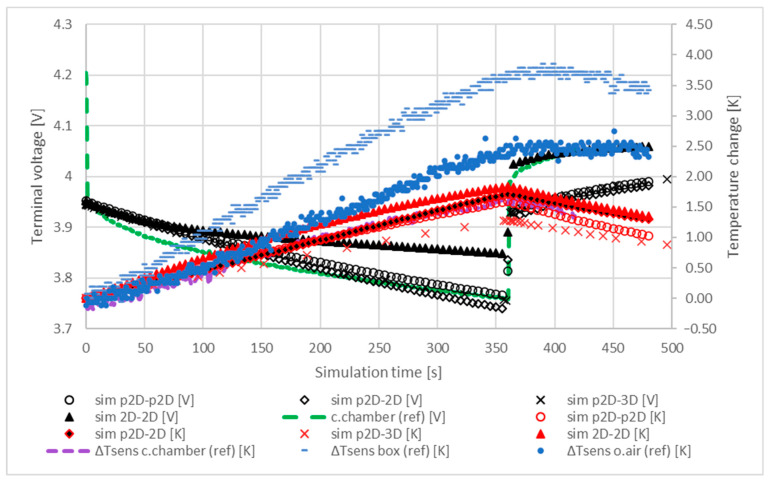
Comparison of the transient response of the electrochemical and thermal models during an interrupted discharge of 1 C. The cell started to relax after 360 s. A rate of heat convection of 0.1 m/s was applied for cooling.

**Figure 8 materials-16-07613-f008:**
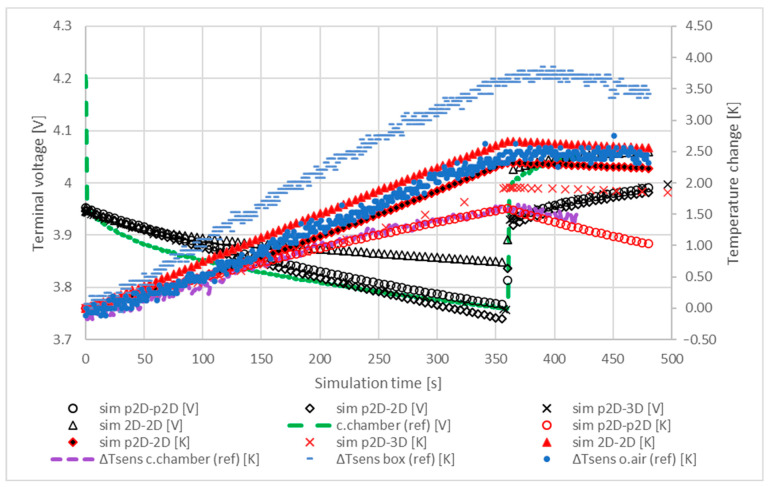
Comparison of the transient responses of the electrochemical and thermal models during an interrupted discharge of 1 C. A heat flux of 10$\;W/m^{2}K$ was applied to model unregulated (i.e., natural) cooling.

**Figure 9 materials-16-07613-f009:**
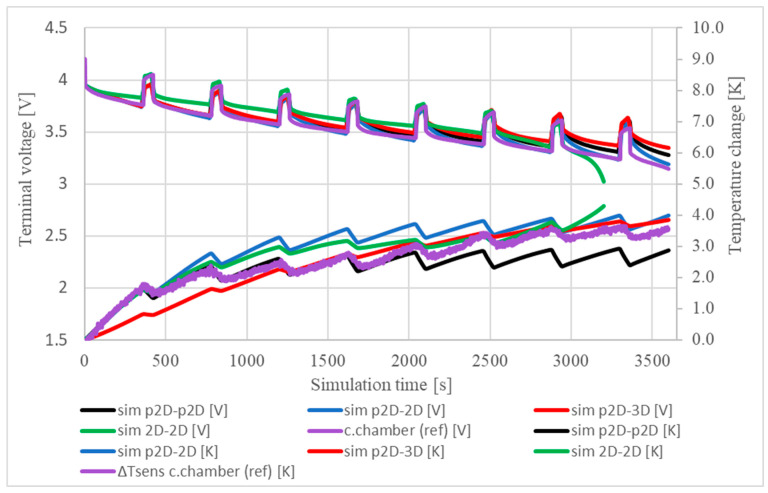
Comparison of the transient responses of the models during 3600 s of simulation. A heat flux of 10 $W/m^{2}K$ was applied to implement cooling.

**Figure 10 materials-16-07613-f010:**
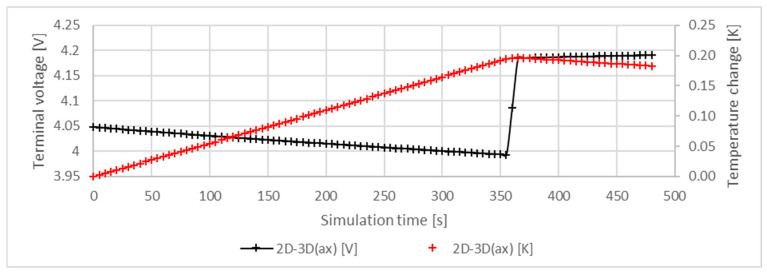
Voltage and temperature characteristics of the 2D-3D(ax) model. A heat flux of 10 $W/m^{2}K$ was applied to model unregulated (i.e., natural) cooling.

**Figure 11 materials-16-07613-f011:**
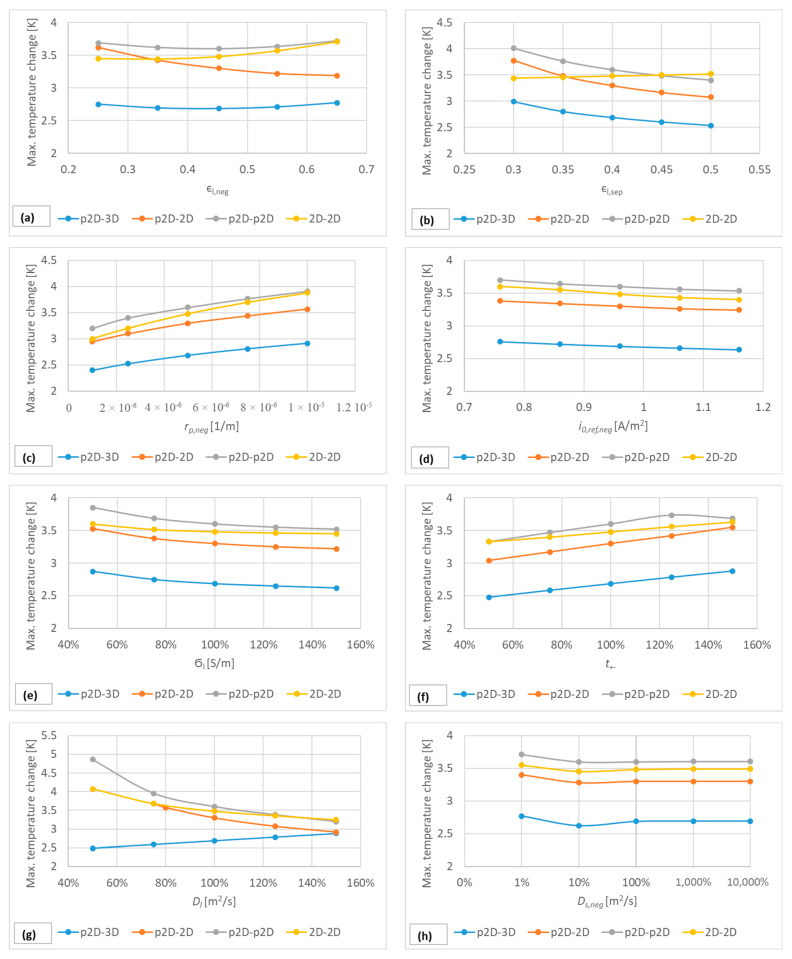
Approximate sensitivity functions of the parameters used in the one-, two- and three-dimensional thermo-electrochemical models that significantly determine battery dynamics. The results were obtained by 5 valuable parameter sweeps, in which the maximum cell temperature was determined for each parameter value. The parameter sweep was performed for all parameters considered. The results for the given parameter can be seen in (**a**–**i**). Note that in (**i**) the heat transfer coefficient was only evaluated in the two-dimensional thermal models.

**Figure 12 materials-16-07613-f012:**
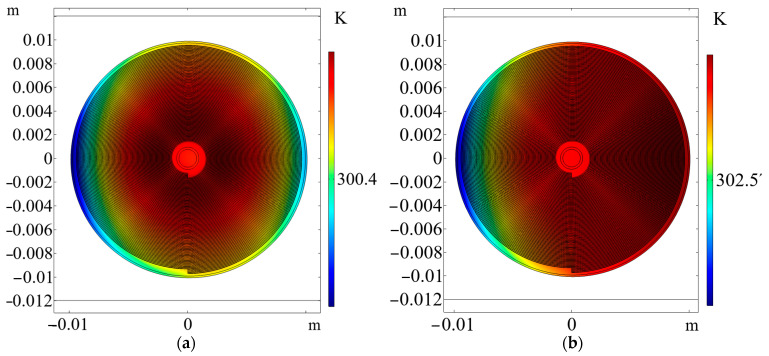
Temperature distribution in the 2D thermal model (**a**) after 1 s and (**b**) after reaching a steady state. The shift in the temperature in (**b**) was due to the heat convection applied which transported heat from left to right.

**Figure 13 materials-16-07613-f013:**
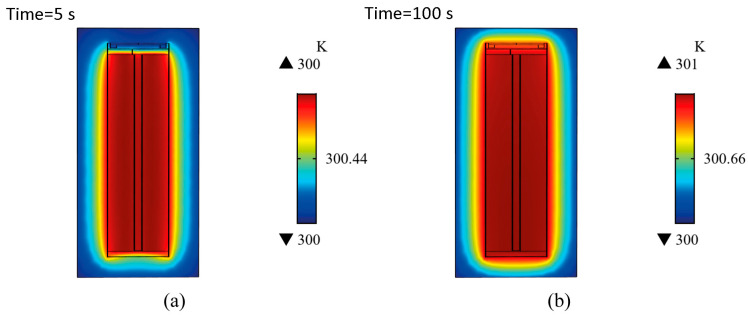
Temperature distribution in the longitudinal section of the 3D thermal model (**a**) after 5 s and (**b**) after 100 s. The effect of the cap, bottom terminal and metal housing was detected. A heat flux of 10 $W/m^{2}K$ was applied to provide a uniform cooling effect on every external boundary.

**Table 1 materials-16-07613-t001:** The governing DFN electrochemical equations in the form required by the simulation environment used in the FEA.

Region	Equation No.	Governing Equation
Charge conservation in solid	(1)	$\epsilon_{s,k}\nabla \cdot {\bar{i}}_{s,k}=\nabla \cdot \left(-\sigma_{eff,k}\nabla \varphi_{s,k}\right)=0$
Charge balance in electrolyte	(2)	$\nabla \cdot {\bar{i}}_{l}=\nabla \cdot \left(-\kappa_{l,eff}\nabla \phi_{l}-\frac{2\kappa_{l,eff}RT}{F}\left(1+\frac{\partial ln\;f_{\pm }}{\partial ln\;c_{l}}\right)\left(t_{+}^{0}-1\right)\nabla \ln c_{l}\right)=0$
Mass conservation in solid	(3)	$\frac{\partial c_{s,k}}{\partial t}=\frac{1}{r^{2}}D_{s,eff,k}\frac{\partial }{\partial r}\left(r^{2}\frac{\partial c_{s,k}}{\partial r}\right)$
Mass conservation in electrolyte	(4)	$\epsilon_{l}\frac{\partial \bar{c_{l}}}{\partial t}=\nabla \cdot \left(D_{l,eff}\nabla \bar{c_{l,k}}\;\right)+a_{k}\left(1-t_{+}^{0}\right)\bar{J}$
Butler–Volmer kinetics	(5)	$\bar{J}=\frac{\bar{i_{0}}}{F}\left(e^{\frac{\left(1-\alpha \right)F}{RT}\bar{\eta }}-e^{-\frac{\alpha F}{RT}\bar{\eta }}\right)$
		*k = n*, *p*, where *n* and *p* represent the anode and cathode, respectively.

**Table 2 materials-16-07613-t002:** The governing thermal equations in the form required by the simulation environment used in the FEA.

Region	Equation No.	Governing Equation
Ohmic losses	(7)	$Q_{Ohmic}=\underset{k}{\sum }(\sigma_{k}^{eff}\nabla \varphi_{s,k}\cdot \nabla \varphi_{s,k})+\underset{k}{\sum }(\kappa_{k}^{eff}\nabla \varphi_{l,k}\cdot \nabla \varphi_{l,k}+\kappa_{D,k}^{eff}\nabla lnc_{l,k}\cdot \nabla \varphi_{l,k})$
Reversible heat losses	(8)	$Q_{rev}=\underset{k}{\sum }a_{s,k}\bar{i_{n,k}}T\left(\frac{\partial U_{k}}{\partial T}\right)\;$
Irreversible heat losses	(9)	$Q_{irrev}=\underset{k}{\sum }a_{s,k}\bar{i_{n,k}}\left(\bar{\varphi_{s,k}}-\bar{\varphi_{l,k}}-U_{k}\right)\;$
Total heat generated	(10)	$Q_{tot}=\sum \rho c\left(\frac{\partial T}{\partial t}\right)+\sum \rho c_{p}v\cdot \nabla T+\sum \nabla \cdot (q_{cond}+q_{rad})$,
		where $q_{cond}=-\lambda_{eff}\cdot \nabla T$ and $q_{rad}=-\sigma A_{cell}\epsilon_{em,cell}\left(T_{cell}^{4}-T_{ambient}^{4}\right)$.
		*k* = *n*, *p*, where *n* and *p* represent the anode and cathode, respectively.

**Table 3 materials-16-07613-t003:** Advantages and disadvantages of the coupled models in different dimensions.

Dimension		Identifying Feature	Advantages	Disadvantages
DFN Model	Thermal Model			
p2D	p2D	Flat geometry model	The best compromise between computational speed and model details.	Missing temperature details in the axial direction and in the fittings.
	2D	Lumped parameter model	The temperature distribution in a radial direction is more realistic.	Missing temperature details in the axial direction and in the cap.
	3D	Lumped parameter model	Temperature distribution in the fittings and each spatial direction is covered. A realistic cooling scenario is easy to implement.	Missing the effect of the layered structure on the heat transport inside the cell.
2D	2D	Spiral geometry model	The most detailed model: the effect of porous electrodes and separator, the voltage drop in the current collectors and the cell capacity defined by the spiral turns are considered.	The high computational demand and meshing is challenging because of the differences in size of several orders of magnitude.
2D	2D-3D (ax)	A 3D model based on axial symmetry	Reduced computational demand due to symmetricity.	The merged electrode structure and the resulting reduced reaction cross-section results in an unrealistic model.

## Data Availability

Data is contained within the article and Appendix A.

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
