# Peer review of "Comparison of Coupled Electrochemical and Thermal Modelling Strategies of 18650 Li-Ion Batteries in Finite Element Analysis—A Review"

_materials, 2023, doi:10.3390/ma16247613_

Round 1

Reviewer 1 Report

Comments and Suggestions for Authors

This paper presents a comparative study on electrochemical-thermal coupled modeling strategies applied to simulate 18650 LIB batteries. The study is comprehensive, offering substantial data to support its claims and holds the potential to serve as a valuable data reference aiding future research in this field. However, to enhance the quality before publication, several concerns should be addressed:

1.      Throughout the paper, there are cross-reference errors that need correction.

2.      As the primary aim is to compare various modeling strategies, it would be beneficial for the authors to add a summary table that lists the average errors of different models in comparison to the experimental data.

3.      In the model development, there are several key parameters that are obtained from literature (such as electrode porosity, particle size, mass transfer coefficients, convection coefficients, etc.). Given that these parameters might not entirely align with the battery tested in this study, it is necessary for the authors to conduct a sensitivity analysis. This analysis would also serve to compare which model strategy performs best concerning parameter uncertainty.

4.      There are several recent relevant studies that are not cited, such as doi.org/10.1021/acs.energyfuels.0c02609.

Author Response

Response in the attachment.

Reviewer 2 Report

Comments and Suggestions for Authors

This paper compares modeling approaches for electrochemical and thermal simulations. A linear cell geometry works well for simple and fast simulations, while a 2D spiral geometry is better for transient simulations, but computationally intensive. Cross-sectional area accuracy is crucial for model performance. 2D geometries offer realistic cross-sectional areas. It is well scructured paper which I suggest to consider for publication, but before the authors should address these point that follow:

Introduction:

  • Correct the subscript in "LiPF6" to make it LiPF₆.
  • The first paragraph lacks proper references; please add relevant citations.
  • When discussing the improved thermal models, provide citations, particularly review papers.
  • Ensure you include references before introducing new concepts or terms.
  • Define "DFN" before using the acronym.
  • Maintain consistency in formatting, whether it's "2C" or "2 C."
  • Define "DoF" in the last line of the introduction.

Scope of the paper:

  • Explain why the study focuses exclusively on 18650 batteries, and if there's a specific rationale beyond their popularity.

Materials and Methods:

  • Modify the title to reflect that it is a review paper, not a research paper.
  • Address the "Error! reference not found" issue and verify all references throughout the text.
  • Align equation 2 properly in Table 1 and add a space after the table.
  • Ensure consistent font size for functions in Chapter 3.1.
  • Align equations in Table 2 and add a space after the table.
  • Clearly indicate the source or origin of Figure 1 and Figure 2, and provide proper references.
  • Relocate "2D" in Figure 2 for clarity.
  • Correct "Area of the separator" to "Area of the separator (A_sep)" in Page 6, line 7.
  • Maintain consistency in capitalization for "D" in Chapter 3.3.
  • Capitalize the first letter of each row in the column "Identifying features" in Table 3.
  • Clarify if it's "p2D" or "2D" in the "Thermal model" column of Table 3.

Experimental:

  • Ensure consistent font size for subscripts, especially between citations 47 and 48.

Results and Discussion:

  • Enhance the visual quality of Figures 9 to 14.
  • Eliminate the gap between Figure 10 and Figure 12.
  • Improve the quality of Figures 15 and 16 or overlay scaling numbers on the axes for better readability. Additionally, provide proper references for these figures in their captions.
  •  

Author Response

Response in the attachment.

Round 2

Reviewer 1 Report

Comments and Suggestions for Authors

I recommend publication. The authors have improved the quality of this manuscript and addressed my concerns.